# Chemical Composition of Essential Oils of Bulbs and Aerial Parts of Two Cultivars of *Allium sativum* and Their Antibiofilm Activity against Food and Nosocomial Pathogens

**DOI:** 10.3390/antibiotics11060724

**Published:** 2022-05-27

**Authors:** Filomena Nazzaro, Flavio Polito, Giuseppe Amato, Lucia Caputo, Rosaria Francolino, Antonio D’Acierno, Florinda Fratianni, Vincenzo Candido, Raffaele Coppola, Vincenzo De Feo

**Affiliations:** 1Institute of Food Sciences, CNR-ISA, Via Roma, 64, 83100 Avellino, Italy; giuseppe.amato119@gmail.com (G.A.); dacierno.a@isa.cnr.it (A.D.); fratianni@isa.cnr.it (F.F.); coppola@unimol.it (R.C.); defeo@unisa.it (V.D.F.); 2Department of Pharmacy, University of Salerno, Via Giovanni Paolo II, 132, 84084 Fisciano, Italy; fpolito@unisa.it (F.P.); lcaputo@unisa.it (L.C.); rfrancolino@unisa.it (R.F.); 3Department of European and Mediterranean Culture, University of Basilicata, Via Lanera, 20, 75100 Matera, Italy; vincenzo.candido@unibas.it; 4Department of Agricultural, Environmental and Food Sciences, University of Molise, 86100 Campobasso, Italy

**Keywords:** *Allium sativum*, chemical composition, EO, biofilm, food pathogens

## Abstract

This work aimed to evaluate the chemical composition of the essential oils (EOs) of two cultivars of *Allium sativum* and their antibiofilm activity against the food pathogens *Acinetobacter baumannii*, *Escherichia coli*, *Listeria monocytogenes*, and *Staphylococcus aureus*. The crystal violet assay ascertained the susceptibility of the bacterial biofilms, while the MTT assay let to evaluations of the metabolic changes occurring in the bacterial cells within biofilms. Their chemical composition indicated some sulfuric compounds (i.e., allicin, diallyl disulfide, and allyl propyl disulfide), and decene as some of the main components of the EOs. The aerial parts and bulbs’ EOs from the two cultivars showed chemical differences, which seemed to affect the antibiofilm activity. The EOs from aerial parts of ‘Bianco del Veneto’ inhibited the biofilm formation of *L. monocytogenes* and *E. coli* (60.55% and 40.33%, respectively). In comparison, the ‘Staravec’ EO inhibited the cellular metabolism of *E. coli* (62.44%) and *S. aureus* (51.52%) sessile cells. These results indicate their possible use as preserving agents in the food industry and suggest their potential exploitation in the development of new formulations to avoid or limit nosocomial infections.

## 1. Introduction

Garlic (*Allium sativum* L.) is one of the most important species used since ancient times as a traditional medicine. Garlic is an annual herbaceous aromatic spice, a member of the Amaryllidaceae family, native to the countries of Central Asia; it was probably one of the first plants to be cultivated and used as food [1,2]. The binomial *Allium sativum* is a name with both Celtic and Latin origins. From the Celtic side, the term ‘allium’ means burning or pungent, while the word ‘sativum’ comes from the Latin side, which means planted or cultivated. On the other hand, the term garlic has Anglo-Saxon origins, with ‘gar-leac’ indicating its flowering stem [3]. There are numerous varieties of garlic, also classified according to the bulb’s color and characterized by different compositions and biological activities [4,5]. Garlic is widely used as a flavoring agent. It possesses various therapeutic properties, and is used in the treatment of earache, whooping cough, colds, and stomach disorders, and in the prevention of cardiovascular diseases [6]. These activities have been confirmed by several clinical and epidemiological trials [7]. Garlic properties derive mainly from organosulfides (allicin, diallyl disulfide, and diallyl trisulfide) and flavonoids (especially quercetin). Several studies have shown that these compounds are responsible for antioxidant and antibacterial activities [8]. Recent in silico studies suggested that *Allium sativum* could act also against SARS-CoV2 infection, although this is not supported by in vivo studies [9,10].

For a long time, *Allium* EOs have been recognized as possessing good antibacterial activity against Gram-positive and Gram-negative strains, particularly *E. coli* [11], *Salmonella*, *Pseudomonas*, and *S. aureus* [12]. Many countries, particularly India, use the garlic EO to treat infectious diseases and prevent food spoilage [13]. Furthermore, hydro-alcoholic extracts of bulbs of *A. sativum* showed antifungal activity, for instance, against clinical isolates of *C. albicans* [14].

In recent years, the expansion of some diseases was correlated to a broadening of the presence, in different environments (for instance, foods, hospitals, and workplaces), of some pathogenic bacteria, such as *Listeria monocytogenes*, *Escherichia coli*, *Staphylococcus aureus*, and *Acinobacter baumannii*. *A. baumannii* exhibited a pattern of multidrug resistance in antibiotic susceptibility tests, and now it is considered as one of the most important Gram-negative pathogenic bacteria just after *P. aeruginosa* [15,16]. *S. aureus*, in particular its methicillin-resistant strains, can cause several nosocomial and community infections. [17]. The entero-hemorrhagic *E. coli* O157:H7 is a worldwide foodborne pathogen causing severe diseases in humans [18]. *L. monocytogenes* is a widespread opportunistic pathogen responsible for the listeriosis foodborne disease. It is a highly occurring pathogen in several western and industrialized countries [19]. These bacteria developed, over the years, stronger evolutionary drug resistance due to their careless use, often in situations where their application was to be considered useless and inappropriate [20]. The higher bacterial resistance to the conventional drugs allowed them to more easily form biofilms, causing a severe problem for food and health [21]. Around 80% of human chronic infections are due to bacterial biofilm formation [22]. Biofilm is the leading source of failure for medical implants, which, infected by bacteria capable of attaching themselves to their surfaces through biofilms, gives rise to higher mortality. Biofilms are also known to cause serious problems in the food industry. In fact, many biofilm-producing bacteria are becoming resistant to the sanitizers used to prevent biofilm formation on foods and food-contact surfaces [23]. The biofilm capacity of bacteria determines their enhanced antibiotic resistance, so that they become refractory to the host’s immune responses, which leads to persistent and recurrent infections. Thus, the interest in natural alternatives to prevent biofilm formation increased the search for natural agents as alternatives to conventional sanitizers, to control the biofilm’s development.

Different papers reported the antibiofilm activity of fresh extracts of *A. sativum* against Shiga-toxi-producing *Escherichia coli* (STEC) [24] and the inhibitory biofilm activity exhibited by aqueous and hydroalcoholic extracts of *A. sativum* against *S. aureus, B. cereus, S. pneumoniae, P. aeruginosa, E. coli* and *Klebsiella pneumoniae* [25]. Recently, the solvent extract of the *A. sativum* bulb acted against the biofilms produced by some bacteria (*Lactobacillus acidophilus, Streptococcus. salivarius, S. mutans,* and *S. aureus*) isolated from patients affected by periodontal and dental caries [26], or against *Streptococcus sanguinis*, involved in the endocarditis [27]. These researches also confirmed how using *Allium* species could be of particular importance given their versatility, acting both in the food sector (as a preserving agent) and for human health (potentially avoiding, for example, the onset of nosocomial infections).

*A. sativum* EO contains several classes of compounds, mainly sulfur compounds, such as allicin, diallyl sulfide, and propyl allyl disulfide, which concur to its biological properties, including the fight against pathogens [28]. Allicin [29], diallyl disulfide [30], and propyl allyl disulfide [8] have antibacterial activities. Among the other non-sulfuric compounds, decene, often detected in *Allium* EO, is known for its inhibitory activity against *Streptococcus mutans* [31]. The chemical composition of a plant depends on many factors, such as the production area, genetic factors, climatic conditions, ripening conditions, and the part of the plant considered. Within a given genus, the chemical composition depends on the species, and within the species, on the cultivar. It, in turn, also contributes to influencing the biological properties, such as the antibacterial and anti-biofilm properties, that a specific plant and its derivatives (extracts, EOs) can then exert. For these reasons, we studied the chemical of the EOs of aerial parts and bulbs of two cultivars of *A. sativum*, the Italian ‘Bianco del Veneto’, and the Albanian ‘Staravec’, which have never been studied before, and their biofilm inhibitory capacity against *A. baumannii*, *E. coli*, *L. monocytogenes*, and *S. aureu*s. In addition, we analyzed their effect on the bacterial cell metabolism within biofilm, which concurs to increase their virulence.

## 2. Results and Discussion

### 2.1. Chemical Composition

The EO yield of the two cultivars was greater from the aerial parts than from the bulbs. In fact, the ‘Bianco del Veneto’ aerial parts furnished 0.16% (*w*/*w*) EO, while the bulb furnished 0.08% (*w*/*w*); in ‘Staravec’, the yields were 0.25% and 0.03%, respectively, for the aerial parts and bulbs. As reported in Table 1, the composition of the EOs between the two cultivars, ‘Bianco del Veneto’ and ‘Staravec’, is quite different. Specifically, in the aerial parts of ‘Bianco del Veneto’, 16 compounds were identified; the major component was allicin (17.5%), followed by decene (17.0%) and allyl propyl disulfide (16.1%). On the other hand, the bulbs have a richer chemical composition, with 59 identified components. Moreover, in this case, the main component was allicin (50.9%), followed by other sulfur compounds such as diallyl disulfide (27.9%), ethyl diallyl trisulfide (3.1%), 2,2-bis(ethylthio)-propane (2.0%), (*Z*)-allyl propyl sulfide (1.4%), and methyl allicin (1.3%).

The chemical composition of the ‘Staravec’ appeared to be more homogeneous between the aerial parts and the bulbs, with 43 and 39 components identified, respectively. In the EO of the aerial parts, the main components were diallyl tetrasulfide (11.4%), allyl propyl disulfide (10.1%), and 12-methyl tridecanoate (8.7%). Allicin accounted only for 0.8%. A different situation was found in the EO from the bulbs, where allicin was the main component, representing 62.2% of the sample. In this EO, other sulfur compounds are present: diallyl disulfide (16.6%) and 2,4-bis(1,1-dimethylphenol) (2,6%).

The presence of numerous sulfur compounds agrees with the available literature [32,33,34,35,36,37]. The amounts of allicin appeared very variable between the samples; in fact, it is present in minimal quantities in the aerial parts of the ‘Staravec’. The other samples represent the main component, with percentages ranging between 17.5% and 62.2%. The presence of sulfur compounds related to diallylsulfide, the main constituents in the EO of the aerial parts of ‘Staravec’, agrees with the available literature. The garlic EO studied by Casella and coworkers [8] is poor in allicin but rich in diallyl disulfide, diallyl trisulfide, and diallyl tetrasulfide. Furthermore, these sulfur compounds influence the biological properties of the samples, including their antibacterial activity, as reported by Shang and coworkers [38].

### 2.2. Antibiofilm Activity

#### Biofilm Inhibitory Capability of the EOs

The capacity of the EOs to inhibit bacterial biofilm formation and the metabolism of the bacterial cells within the biofilm was assessed through crystal violet and MTT tests, respectively. The tests were performed utilizing 10 µL/mL and 20 µL/mL of EOs, two concentrations abundantly lower than the minimal inhibitory concentration, calculated through the resazurin test (Table 2). The data indicated that the bulbs of ‘Bianco del Veneto’ exhibited inhibitory activity on biofilm formation (Table 3), with inhibition percentages ranging from 18.59% (10 µL/mL vs. *A. baumannii*) up to 63% (64.29% with 20 µL/mL vs. *L. monocytogenes* and 63.18% vs. *S. aureus*). The ‘Bianco del Veneto’ EO aerial parts were effective against all four strains, mainly when used at 20 µL/mL, showing an efficacy, especially against *E. coli* and *L. monocytogenes*, against which its inhibitory action reached percentages of 60.55% and 50.52%, respectively. Concurrently, the ‘Bianco del Veneto’ bulbs’ EO was always active, with an inhibitory action ranging between 48.90% (against *E.coli*) up to 64.29% (against *L. monocytogenes*), when used at 20 μL/mL. The EOs from the aerial parts of ‘Staravec’ showed less inhibitory activity, resulting in effectiveness against *A. baumannii* (45.61% inhibition), *E. coli* (27.56% inhibition), and *S. aureus* (26.31% inhibition), but it was utterly ineffective vs. *L. monocytogenes*. The EOs of the bulbs, on the other hand, proved to be extremely ineffective in inhibiting the formation of biofilm by all four microbial strains.

The correlation analysis, considering the chemical composition of the EOs and their inhibitory behavior exhibited against the bacterial biofilm formation, highlighted a different influence of some among the more abundant compounds. Thus, only diallyl sulfide, present in the EO of the aerial parts of the ‘Bianco del Veneto’, seemed to have some inhibitory activity vs. *A. baumannii* (Corr-Coeff = 0.142). Considering the ‘Bianco del Veneto’ bulbs’ EO, where the resulting concentration of diallyl sulfide was equal to 27.9%, such actions arrived up to a concentration of 57.34% using 20 μL/mL of the same oil. The diallyl sulfide, therefore, seemed to counteract the negative effect on the inhibitory biofilm activity exhibited by the decene (Corr-Coeff = −0.19), allyl propyl disulfide (Corr- Coeff = −0.01), and allicin (Corr-Coeff = −0.377). As evidence of the negative effect exerted by allicin, we could highlight how the EO obtained from the ‘Staravec’ bulb’s EO—containing 62.2% of allicin—was unable to exert any inhibitory biofilm effect on all four bacteria used in the tests. The inhibitory biofilm action exerted by the ‘Bianco del Veneto’aerial parts’ EOs against *E. coli* seemed to depend on the decene (Corr-Coeff = 0.488) and allyl propyl disulfide (Corr-Coeff = 0.42). In the case of the ‘Bianco del Veneto’ bulbs’ EO, the good inhibitory biofilm action against *E.coli* seemed to be influenced not so much by the decene (absent in this EO) but rather by the diallyl sulfide. As we have seen previously, the EOs obtained from the ‘Staravec’ cultivar were much less effective in exhibiting inhibitory biofilm action. The correlation analysis indicated that allyl propyl disulfide affected the inhibitory biofilm formation shown by the aerial parts’ EO (whose activity did not go beyond 27.56% with 20 μL/mL). Moreover, as we previously indicated, the EO of the ‘Staravec’ bulb, where allicin (Corr-Coeff = −0.383) is present at 62.2%, was practically ineffective vs. *E. coli*. The correlation coefficients were all positive in the case of *L. monocytogenes*. However, as shown in Table 3, the EOs extracted from the ‘Staravec’ proved utterly ineffective. Diallyl sulfide (Corr-Coeff = 0.493) was present at a negligible percentage (0.5%); the ineffectiveness of the inhibitory biofilm action exhibited by the ‘Staravec’ bulbs’ EO seemed, in this case, to be influenced by the extremely high percentage of allicin (Corr-Coeff = 0.075).

From the analysis of the correlation, the inhibitory biofilm action (up to 63.18%, see Table 3) exhibited by the EO of the bulb of ‘Bianco del Veneto’ against *S.aureus*, seemed to be influenced not so much by allicin, although it was abundant (Corr-Coeff = 0.018), but from the diallyl sulfide (Corr-Coeff = 0.532). Conversely, the positive influence of diallyl sulfide present in the aerial parts’ EO of such a cultivar seemed to be somewhat overwhelmed by the presence of allicin, but above all by the presence of allyl propyl disulfide (Corr-Coeff = −0.305), which appeared to weakly inhibit the *S. aureus*’s biofilm formation.

The available literature demonstrated that diallyl sulfides exhibited promising antimicrobial activity [37,39,40,41,42]. Our results confirmed that, although allicin contributes to the antibacterial properties of hydroalcoholic extracts of garlic [43,44], it could exhibit a breaking effect on the biofilm inhibitory action of the *A. sativum* EOs.

Through the test with MTT, we observed that the action of EOs on the formation of bacterial biofilms did not always coincide with a similar effect on the metabolism of bacterial cells (Table 4).

The action of the EOs on the metabolism of *L. monocytogenes* appeared opposite to what we saw in the test with crystal violet: in fact, the ‘Bianco del Veneto’ EOs, very effective in limiting the adhesion process of the biofilm by this bacterium, were utterly ineffective in affecting its metabolism. On the other hand, the ‘Staravec’ EOs confirmed their almost entirely ineffectiveness (apart from a 10.38% metabolic inhibition in the presence of 20 µL/mL of the EO obtained from the aerial parts). In this case, the correlation analysis highlighted a robust negative effect linked to the presence of allicin (Corr-Coeff = −0.748) and diallyl sulfide (Corr-Coeff = −0.724). The weak albeit inhibitory activity (10.38%) observed in the presence of 20 µL/ mL of the EO obtained from the aerial parts of ‘Staravec’ could be due to the presence of allyl propyl disulfide (Corr-Coeff = 0.284). The EOs’ components might be able to act against its biofilm formation not so much by working on *L. monocytogenes* metabolism but with other mechanisms, as has been reported [45].

Against *E. coli*, an inhibitory effect on biofilm formation (50.52% with the EOs from the aerial parts of ‘Bianco del Veneto’) did not correspond to an identical inhibitory force on the metabolism of these bacterial cells, which was utterly absent. On the contrary, ‘Starevec’ EOs proved to be very efficient at affecting its metabolism, so much so that we observed percentages of metabolic inhibition never lower than 62.64% and 65.71% with 20 µL/mL of the bulb and aerial parts EOs, respectively.

The EOs, when used at the highest concentration, always exerted an inhibitory force on the bacterial metabolism of *A. baumannii*, being that the ‘Staravec’ EOs are more effective than the ‘Bianco del Veneto’ ones, especially in regards to the EO recovered from the bulbs (59.79% and 24.10%, respectively, when we tested 20 µL/mL). Correlation analysis allowed us to observe that allyl propyl disulfide (Corr-Coeff = 0.244) and allyl propyl disulfide (Corr-Coeff = 0.16) seemed to mitigate, in a certain sense, the negative effect of allicin (Corr-Coeff = −0.178) on the inhibitory metabolic capacity exerted by the ‘Bianco del Veneto’ aerial parts’ EOs vs. *A. baumannii*. The lack of decene (Corr- Coeff = 0.553) in the EO of the ‘Bianco del Veneto’ bulb made it relatively weak in inhibiting the *A.baumanni* metabolism. Therefore, we could hypothesize that the greater inhibitory metabolic capacity exerted by the EO of the aerial parts of ‘Staravec’ vs. *A. baumannii* could be due to the almost total absence of allicin. On the other hand, the bulb EO of ‘Staravec’ showed a greater propensity to inhibit the metabolism of *A. baumannii*, probably due to allyl propyl disulfide and decene, expressing a positive correlation coefficient, although both are present at low concentrations.

The bacterial metabolism of *S. aureus* was, in turn, the most sensitive to the action of all the EOs tested. 20 µL/mL of the ‘Bianco del Veneto’ bulbs’ EOs inhibited the adhesion process and bacterial biofilm formation (63.18% inhibition, Table 3). In addition, it exerted a potent inhibitory force on its cellular metabolism (61.44% inhibition).

Correlation analysis evidenced a strong influence exhibited by allicin (Corr-coeff = 0.449), which, on the other hand, was also the most abundant compound of these two EOs. The inhibitory effect of allicin on the *S. aureus* metabolism was weakened by the presence of allyl propyl disulfide (Corr-coeff = −0875) and decene (Corr-Coeff = −0.928). The different behavior exhibited by the EOs on the biofilm adhesion process and the metabolism of bacterial cells confirms previous studies on other *Allium* species and cultivars [28,46]. Casella and coworkers [8] demonstrated that the antibacterial activity of the EOs from different *Allium* species’ fresh bulbs was mainly due to diallyl sulfides. This was partially ascertained by Polito et al. [28]. Taking into account the results of their work and calculating the correlation coefficients, we could hypothesize an antagonistic activity of allicin on the capacity of the ‘Rosso di Sulmona’ and ‘Rosso di Spagna’ EOs to inhibit biofilm formation (Corr-Coeff for *A.baumannii* = −0.095 Corr-Coeff for *E.coli*: −0.644; Corr-Coeff for *L. monocytogenes* = −0.013; Corr-Coeff for *S.aureus* =-0.314). Concurrently, diallyl disulfide positively influenced the inhibitory biofilm capacity of those EOs against *E,coli* (Corr-coeff = 0.810), while propyl allyl disulfide influenced the inhibitory ability of those EOs against *S.aureus* (Corr-Coeff = 0.767) and *L. monocytogenes* (Corr-Coeff = 0.474).

On the other hand, allicin positively influenced the capacity exhibited by those EOs to inhibit the metabolism of *A. baumannii* within the biofilm (Corr-coeff = 0.850). Thus, performing the correlation analysis based on the data reported in the work of Polito et al. [28], we might propose an almost generally negative influence of allicin and a positive effect exhibited generally by other sulfide compounds, such as diallyl sulfides and allyl propyl sulfides. In addition, other compounds, such as decene, are similar in their capacity to inhibit the formation of biofilms or the cell metabolism within the biofilm.

Yin and Cheng [39] reported some properties of diallyl sulfides, demonstrating their capacity to offer antioxidant and antimicrobial protection against pathogens, including *E. coli, L. monocytogenes*, and *S. aureus*. These compounds also acted on some bacterial enzymes involved in bacterial metabolism [47]. The EOs obtained from the two cultivars ‘Bianco del Veneto’ and ‘Staravec’, could inhibit biofilm formation and, in some cases, such as against *E. coli* and *L. monocytogenes*, were capable of affecting their metabolism. However, Fratianni et al. [48], analyzing the action of different types of honey on some pathogenic bacteria, observed that the inhibitory effect on biofilm formation is not always correlated with an analog effect on the bacterial metabolism within the biofilm. Similarly, the action of different seed oils against *A. baumannii*, *E. coli*, *L. monocytogenes*, *P. aeruginosa*, and *S. aureus* was sometimes different, depending on the microorganism [49].

## 3. Materials and Methods

### 3.1. Plant Material

Plants of the two cultivars of *A. sativum*, ‘Bianco del Veneto’ and ‘Staravec’, were collected in May–June 2020. The cultivars were grown in an experimental field at Pontecagnano (Salerno province, Southern Italy) on a fine-textured soil previously ploughed and fertilized. Cloves of both cultivars were planted on 15 November 2019, spaced 10 cm apart from each other and 10 cm deep in the soil, in rows spaced 50 cm apart, to obtain a density of 20 plants per square meter. According to a randomized block design with 3 reps, the 2 cultivars were arranged in 5 m^2^ plots (2.0 m × 2.5 m). Furthermore, the standard agronomic practices of local garlic growers were followed. At harvest time, samples of 10 plants, randomly taken from each plot, were analyzed for morphological traits: the skin color of the bulbs and cloves, bulbs’ mean weight, equatorial bulb diameter, number of cloves per bulb, cloves’ mean weight. As shown in Table 5, the bulb and clove traits of garlic were significantly different between the cultivars. In particular, the mean weight and equatorial diameter of the ‘Bianco del Veneto’ bulbs were significantly higher than the ‘Staravec’ ones. In addition, the mean clove weight of ‘Bianco del Veneto’ was higher, whilst the number of cloves per bulb was significantly similar between the tested garlic cultivars.

### 3.2. Extraction of the EOs

Plant samples were cleaned from soil and other material residues and dried for about one week at room temperature. The plant material was divided into aerial parts and bulbs, separated and classified, and extracted with methanol at room temperature. This extraction was repeated three times, renewing the solvent. The extracts were then filtered using a paper filter and freed of excess methanol (Rotavapor R-100, BUCHI Italia s.r.l, 20007 Cornaredo, Italy). Subsequently, with the minimum amount of methanol, samples were placed in a flask half-filled with water and subjected to steam distillation, as reported in the European Pharmacopoeia [50]. The EOs obtained were solubilized in n-hexane, dried in a nitrogen atmosphere, and stored in amber vials in the refrigerator at 4 °C.

### 3.3. Composition of the EOs

The composition of the EOs was achieved by GC and GC-MS. GC analysis was performed using a Sigma 115 gas chromatograph (Perkin-Elmer Italia, Milano, Italy) equipped with a flame ionization detector (FID) and a non-polar HP-5 MS fused silica capillary column (30 m × 0.25 mm; 0.25 μm film thickness). Injector and detector temperatures were 250 °C and 290 °C, respectively. The analysis was carried out as follows: 5 min isothermally at 40 °C; the temperature was subsequently increased by 2 °C/min until 270 °C; and finally, the sample was kept in isotherm for 20 min. The analysis was also performed on an HP Innowax column (50 m × 0.20 nm; 0.25 μm film thickness). In both cases, He was used the carrier gas at a flow of 1.0 mL/min. GC-MS analysis was performed using a 6850 Ser II Apparatus (Agilent, Cernusco sul Naviglio, Italy) equipped with a DB-5 fused silica capillary column (30 m × 0.25 mm; 0.25 μm film thickness) and linked to an Agilent Mass Selective Detector (MSD 5973). Ionization voltage 70 V; ion multiplier energy 2000 V. The mass spectra were scanned in the range of 40–500 amu, with 5 scans per second. The chromatographic conditions were as above; the transfer line temperature was 295 °C. Most of the components were identified by comparison of their Kovats indices (Ki) with those of the literature [51,52,53] and by comparison of the mass spectra to those of pure compounds available in our laboratory or to those present in the NIST 02 and Wiley 257 mass libraries. The Kovats indices were determined in relation to a homologous series of *n*-alkanes (C10–C35), under the same operating conditions. For some compounds, the identification was confirmed by co-injection with standard compounds.

### 3.4. Antibacterial Activity

#### Microorganisms and Culture Conditions

The Gram-positive, *Staphylococcus aureus* subsp. *aureus* (ATCC 25923) and *Listeria monocytogenes* (ATCC 7644), and the Gram-negative, *Escherichia coli* (DSM 8579) and *Acinetobacter baumannii* (ATCC 19606), were cultured in Luria–Bertani broth at 37 °C and 80 rpm (Corning LSE, Pisa, Italy) for 18 h (*A. baumannii* was cultured at 35 °C).

### 3.5. Minimal Inhibitory Concentration (MIC)

The resazurin microtiter plate assay was used to determine the lowest concentration of the EOs capable of preventing visible growth of the microorganisms [54]. Multiwell plates were organized in triplicate; then, they were incubated at 37 °C and 35 °C, depending on the strain, for 24 h. The smallest concentration at which a colour change happened (from dark purple to colourless) fixed the MIC value.

### 3.6. Biofilm Inhibitory Action of the EOs

The EOs’ capability to inhibit bacterial biofilm formation was assessed in flat-bottomed 96-well microtiter plates [46], using two concentration of EOs lower than those detected through MIC, which is applied generally to identify the concentration of a substance/molecule capable of inhibiting the growth of the planktonic cells. A total of 10 µL of the overnight bacterial cultures, standardized to 0.5 McFarland (1.5 × 10^7^ cells/mL) with fresh culture broth, were put in each well. Then, 10 and 20 µL/mL of each EO and Luria–Bertani broth were added, to reach a final volume of 250 µL/well. Microplates were closed with parafilm, to prevent evaporation; they were incubated for 48 h at 37 °C (except for *A. baumannii*, incubated at 35 °C). Once removed the planktonic cells, wells were washed with sterile PBS, and we added 200 µL of methanol to fix the adhered cells. After 15 min, the methanol was removed, and the microplates were left to dry. The staining of the adhered cells was obtained by adding 200 µL of 2% *w*/*v* crystal violet solution, which was removed after 20 min. The wells were gently washed with sterile PBS and left to dry under the flow laminar cap. The addition of 200 µL of glacial acetic acid 20% *w*/*v* allowed for the release of the bound dye. The absorbance was measured at λ = 540 nm (Cary, Varian, Milano, Italy). The percent of adhesion was calculated with respect to the control; an inhibition of 0% was considered for cells without treatment. The tests were carried out in triplicate, and the average results were taken for reproducibility.

### 3.7. Inhibition of Cells’ Metabolic Activity within the Biofilm

The EOs capacity to inhibit the metabolic activity of the bacterial cells was assessed through the 3-(4,5-dimethylthiazol-2-yl)-2,5-diphenyltetrazolium bromide (MTT) method [55]. The method is different from MIC, as it is applied to the cells within the biofilm after the discarding of the non-adhered cells; furthermore, it was used considering concentrations of the EOs lower than those identified by MIC. Ten µL of the overnight bacterial cultures, standardized to 0.5 McFarland (1.5 × 10^7^ cells/mL) with fresh culture broth, were put in each well. Then, 10 and 20 µL/mL of each EO and Luria–Bertani broth were added to reach a final volume of 250 µL/well. After 48 h of incubation, planktonic cells were removed, and 150 µL of PBS and 30 µL of 0.3% of MTT (Sigma, Milano, Italy) were added. Depending on the strain, microplates were kept at 37 °C and 35 °C. After 2 h, we removed the MTT solution, and two washing steps were performed with 200 µL of sterile physiological solution. Then, 200 µL of dimethyl sulfoxide (Sigma, Milano, Italy) was added to allow for the dissolution of the formazan crystals by the adhered cells, which was measured after 2 h, at λ = 570 nm (Cary, Varian, Milano, Italy).

### 3.8. Statistical Analysis

All assays were carried out in triplicate. Data from each experiment were expressed as the mean ± SD, and statistically analyzed by two-way ANOVA followed by Dunnet’s multiple comparisons test, at the significance level of *p*
*<* 0.05, using GraphPad Prism 6.0. The analysis also correlated the values of the antibiofilm activity of the EOs with their composition, using the free software environment for statistical computing and graphics, R (https://www.r-project.org/ last accessed on 18 May 2022).

## 4. Conclusions

This research demonstrated a chemical difference between the EOs of two cultivars of *A. sativum*. The chemical compositions confirmed the presence of sulfur compounds, particularly allicin, as reported in the literature. However, the percent amounts of allicin were variable between the two cultivars and between the aerial parts and bulbs of the same cultivar; indeed, allicin, rarely affected the inhibitory biofilm capabilities exhibited by EOs. Both the EOs from the aerial parts and the bulbs of the cv. ‘Bianco del Veneto’, unlike the cv. ‘Staravec’, were able to inhibit the biofilm formation of *L. monocytogenes*. On the other hand, the EOs from aerial parts bulbs of ‘Staravec’ acted against the cellular metabolism of *E. coli*.

Identifying substances with a broad spectrum of antibacterial activity against Gram-positive and Gram-negative bacteria, and their virulence, is always helpful from an application point of view, both for food safety and for different areas such as hospitals.

The inhibitory action exhibited not only by the bulbs but also by the aerial parts of the two cultivars of *A. sativum* represents an aspect of particular interest. These results could allow for the more rational use and exploitation of all the *A. sativum* plants. Most people continue to harvest bulbs exclusively and produce a large amount of *Allium* plant waste, which could have important biological properties. This might give several advantages to the food industry, as the extracts/EOs of the aerial parts could be helpful in the formulation of natural preserving agents to prolong the shelf life of perishable products. Furthermore, other studies on these substances could give the industry new weapons to fight pathogenic bacteria that are becoming more easily resistant to conventional antibiotic therapies. Finally, as we have tried to highlight, the results show once again that the biological properties of a plant are the result of the synergistic or antagonistic actions of the singular substances that compose it. Biodiversity is also like this, which could provide a full advantage to the circular economy, which today represents the most crucial pillar from an environmental, economic, and social point of view.

## Figures and Tables

**Table 1 antibiotics-11-00724-t001:** Chemical composition the EOs from the *A. sativum* cultivars ‘Bianco del Veneto’ and ‘Staravec’.

N.	Compound	%				RT	KI
		‘Bianco del Veneto’		‘Staravec’			
		Aerial Part	Bulbs	Aerial Part	Bulbs		
1	2,4-Dimethylhexane	-	0.1	1.4	0.2	5.1	758
2	Methyl-2-propenyl disulfide	-	T	-	-	8.0	797
3	2,2-Bis(ethylthio)-propane	-	2.0	-	0.4	9.2	812
4	α-Pinene	-	-	0.2	-	9.8	819
5	2,2-Dimethylhexyl propanoate	-	T	-	-	9.8	819
6	2,2-Dimethylbutyl propanoate	-	T	-	-	10.2	824
7	2,4-Dimethyldecane	-	T	0.5	T	10.9	834
8	2,2,3,4-Tetramethylpentane	-	T	-	-	11.1	836
9	2,6-Dimethylnonane	-	T	-	T	11.2	837
10	1,8-Cineole	-	T	0.1	T	11.4	841
11	2,3,5,8-Tetramethyldecane	-	0.1	1.5	0.2	12.2	851
12	4-Methyl-1-undecene	-	T	0.6	1.8	12.4	853
13	2-Hydroxyethyl disulfide	0.9	0.1	-	0.2	15.2	889
14	3,4-Dimethyl thiophene	-	-	0.2	T	15.3	891
15	Bis (1,1-dimethylpropyl) disulfide	-	-	0.7	-	16.4	896
16	Tridecane	-	0.1	-	0.1	16.6	907
17	1,11-Thio-bis-butine	-	-	1.0	-	16.7	909
18	Dimethyl sulfide	-	0.8	0.9	0.7	17.7	921
19	(*Z*)-Methyl propenyl disulfide	-	-	0.8	-	18.0	925
20	2,6,10-Trimethyldodecane	-	-	1.4	0.2	18.2	928
21	Dodecylsulfide	-	-	0.2	T	18.5	932
22	Dodecyl-7-en disulfide	-	-	3,0	0.8	18.6	934
23	Dodecyl-8-en disulfide	-	-	2.1	-	18.9	937
24	2-Methoxy-4-vinylphenol	11.1	-	2.4	-	19.9	951
25	Hexanal	-	-	0.6	-	20.1	953
26	(*E*)-Allyl propyl sulfide	-	0.3	0.4	0.4	21.1	965
27	(*Z*)-Allyl propyl sulfide	-	1.4	1.4	-	21.6	972
28	Hexanol	-	-	0.8	-	22.2	979
29	Octane	-	-	1.8	1.7	22.4	982
30	Decane disulfide	-	-	1.5	-	23.1	992
31	Geranyl isovalerate	-	-	0.5	-	23.3	994
32	Nonanal	-	-	3.9	-	23.4	995
33	Nonene	-	-	4.0	-	24.1	1000
34	Decene	17.0	-	8.2	0.2	24.3	1003
35	2,4-Bis (1,1-dimethylphenol)	0.8	0.9	1.3	2.6	24.7	1009
36	2-Butyloctanol	-	0.9	3.6	0.8	25.1	1014
37	3,7,12-Trimethyldodecan-1-ol	0.6	-	-	-	25.3	1017
38	(E)-9-Octadecene	-	0.1	0.2	0.1	26.5	1033
39	4-Methylundecene	0.8	0.1	-	-	26.6	1035
40	*n*-Nonane	-	-	1.5	-	26.8	1038
41	Propyl trisulfide	-	0.3	1.6	-	27.4	1046
42	1,3,5-Tritiane	9.4	0.9	3.0	-	28.3	1057
43	Undecane	-	-	1.1	-	28.4	1057
44	Undecene	-	-	1.2	-	28.6	1061
45	Methyl 2-propenyl trisulfide	1.9	-	-	-	28.9	1065
46	12-Methyl tridecanoate	-	0.3	8.7	-	29.5	1073
47	Methyl triacontanoate	-	0.1	0.8	0.2	29.6	1075
48	Ethyl 2-oxo-tetradecanoate	-	0.1	0.9	0.2	30.0	1080
49	Propenyl trisulfure	-	0.2	4.7	0.7	30.0	1081
50	2-Butyl-2-ethylpropanediol	-	0.1	-	-	30.5	1086
51	Methyl pentadecanoate	7.0	0.7	-	0.6	30.8	1091
52	12-Methyl tetradecanoate	-	0.3	-	0.4	31.0	1093
53	2-Hexyloctanol	-	0.3	-	-	31.1	1095
54	Vinyl trisulfide	-	-	-	0.4	31.9	1099
55	2-Butyloctanol	0.7	-	-	-	32.0	1101
56	14-Methyl pentadecanoate	-	0.4	-	0.4	32.8	1113
57	9-Methyl esadecanoate	0.6	1,0	-	0.2	33.1	1116
58	Diallyl disulfide	8.1	27.9	0.5	16.6	33.8	1126
59	Allyl propyl disulfide	16.1	0.1	10.1	1.1	34.1	1131
60	14-Methyl esadecanoate	0.3	0.4	-	-	35.0	1143
61	Methyl heptanoate	-	0.5	0.2	0.4	35.5	1151
62	9- Methyl octadecenoate	-	T	-	-	36.2	1160
63	Allicin	17.5	50.9	0.8	62.2	36.9	1171
64	Methyl allicin	5.3	1.3	-	0.7	37.3	1177
65	(*Z*)-Hexadecenal	-	0.1	-	0.1	37.5	1179
66	Methyl octadeca-8,11-dienoate	-	1.3	-	1.2	37.7	1182
67	Methyl-10-oxo-octadecanoate	-	0.2	-	0.1	38.0	1187
68	Diallyl trisulfide	-	0.3	-	0.3	38.2	1189
69	Methyl diallyl trisulfide	-	1.1	-	0.9	38.4	1192
70	Ethyl diallyl trisulfide	-	3.1	-	0.6	39.6	1203
71	Vinyl diallyl trisulfide	-	0.1	-	-	40.8	1221
72	(*Z*)-7-Hexadecenal	-	0.1	-	-	41.8	1238
73	Di-*tert*-dodecyl disulfide	-	T	-	-	42.1	1242
74	5,9,13-Trimethyl tetradecanoate	-	0.1	-	-	42.4	1246
75	Methyl esacosanoate	-	T	-	-	43.9	1270
76	Diallyl tetrasulfide	-	-	11.4	-	48.4	1338
	Total	98.1	99.1	92.8	97.7		

RT = retention time; KI = Kovats index on an HP5 MS capillary column; T = traces, less than 0.05%; - = absent.

**Table 2 antibiotics-11-00724-t002:** Minimal inhibitory concentration (µL/mL) of the EOs from the *A. sativum* cultivars necessary to inhibit the growth of the pathogenic bacterial strains *Listeria monocytogenes, Escherichia coli, Acinetobacter baumannii, Staphylococcus aureus*.

		*A. baumannii*	*E. coli*	*L. monocytogenes*	*S. aureus*
‘Bianco del Veneto’	Aerial Parts	40 ± 2	30 ± 3	30 ^b^ ± 3	40 ± 2
	Bulbs	30 ± 2	30 ± 3	30 ^b^ ± 3	30 ^b^ ± 2
‘Staravec’	Aerial Parts	30 ± 2	35 ± 3	40 ± 2	35 ± 3
	Bulbs	40 ± 2	40 ± 4	40 ± 2	40 ± 3
Tetracycline		31 ± 1	24 ± 3	39 ± 2	38 ± 2

The experiments were performed in triplicate and reported as the mean (±SD). ^b^: *p* < 0.01; compared with the tetracycline used as the control (ANOVA followed by Dunnett’s multiple comparison test).

**Table 3 antibiotics-11-00724-t003:** Inhibitory activity of the EOs from the *A. sativum* cultivars on the biofilm formation capacity by the pathogenic bacterial strains *Acinetobacter baumannii*, *Escherichia coli*, *Listeria monocytogenes*, *Staphylococcus aureus*.

		*A. baumannii*	*E. coli*	*L. monocytogenes*	*S. aureus*
‘Bianco del Veneto’	Aerial parts 10 µL/mL	0	17.63 ^a^ ± 1.77	47.23 ^a^ ± 0.84	0
	Aerial parts 20 µL/mL	25.18 ^a^ ± 3.79	50.52 ^a^ ± 1.61	60.55 ^a^ ± 1.30	16.70 ^a^ ± 1.14
	Bulbs 10 µL/mL	18.59 ^a^ ± 2.5	40.31 ^a^ ± 1.46	59.18 ^a^ ± 0.54	49.69 ^a^ ± 1.19
	Bulbs 20 µL/mL	57.34 ^a^ ± 1.34	48.90 ^a^ ± 1.97	64.29 ^a^ ± 1.77	63.18 ^a^ ± 1.15
‘Staravec’	Aerial parts 10 µL/mL	18.09 ^a^ ± 0.34	5.06 ^a^ ± 0.16	0	11.25 ^a^ ± 1.4
	Aerial parts 20 µL/mL	45.61 ^a^ ± 0.16	27.56 ^a^ ± 0.12	0	26.31 ^a^ ± 0.9
	Bulbs 10 µL/mL	0	0	0	0
	Bulbs 20 µL/mL	0	0	0	0

The tests were performed using 10 µL/mL and 20 µL/mL. All tests were performed in triplicate. Results are expressed as percentages (mean ± SD) and calculated assuming the control (untreated bacteria, for which we assumed an inhibitory value = zero). ^a^: *p* < 0.1; compared with the control (ANOVA followed by Dunnett’s multiple comparison test).

**Table 4 antibiotics-11-00724-t004:** Inhibitory activity of the Eos from the *A. sativum* cultivars on the cell metabolism of the *Acinetobacter baumannii*, *Escherichia coli*, *Listeria monocytogenes*, *Staphylococcus aureus* within the biofilm.

		*A. baumannii*	*E. coli*	*L. monocytogenes*	*S. aureus*
‘Bianco del Veneto’	Aerial parts 10 µL/mL	19.73 ^a^ ± 0.17	0	0	0
	Aerial parts 20 µL/mL	45.19 ^a^ ± 0.84	0	0	12.77 ^a^ ± 1.14
	Bulbs 10 µL/mL	0	0	0	47.32 ^a^ ± 1.19
	Bulbs 20 µL/mL	24.10 ^a^ ± 1.66	0	0	61.44 ^a^ ± 1.15
‘Staravec’	Aerial parts 10 µL/mL	46.86 ^a^ ± 1.46	62.44 ^a^ ± 1.34	0	3.91 ^a^ ± 0.13
	Aerial parts 20 µL/mL	54.13 ^a^ ± 2.96	65.71 ^a^ ± 1.70	10.38 ^a^ ± 0.95	52.71 ^a^ ± 1.17
	Bulbs 10 µL/mL	0	45.74 ^a^ ± 1.81	0	51.52 ^a^ ± 1.73
	Bulbs 20 µL/mL	59.79 ^a^ ± 1.12	62.64 ^a^ ± 0.83	0	55.55 ^a^ ± 1.27

The tests were performed using 10 µL/mL and 20 µL/mL. All tests were performed in triplicate. Results are expressed as percentages (mean ± SD) and calculated assuming the control (untreated bacteria, for which we assumed an inhibitory value = zero). ^a^: *p* < 0.1; compared with the control (ANOVA followed by Dunnett’s multiple comparison test).

**Table 5 antibiotics-11-00724-t005:** Morphological traits of bulbs and cloves of *A. sativum* cultivars.

Cultivars ^1^	Bulb Skin Color	Clove Skin Color	Bulb Mean Weight	Bulb Equatorial Diameter	Cloves Per Bulb	Clove Mean Weight
			(g)	(mm)	(n.)	(g)
‘Bianco del Veneto’	white	white	47.2 ± 1.16 ^a^	53.1 ± 0.74 ^a^	13.2 ± 0.75 ^a^	3.0 ± 0.10 ^a^
‘Staravec’	white	white	40.0 ± 0.96 ^b^	47.8 ± 1.16 ^b^	12.9 ± 0.60 ^a^	2.6 ± 0.06 ^b^

^1^ Means followed by the same letters in the same column are not significantly (*p* ≤ 0.05) different.

## Data Availability

The original contributions presented in the study are included in the article. Further inquiries can be directed to the corresponding author.

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
