# Peer review of "Chemical Composition of Essential Oils of Bulbs and Aerial Parts of Two Cultivars of Allium sativum and Their Antibiofilm Activity against Food and Nosocomial Pathogens"

_antibiotics, 2022, doi:10.3390/antibiotics11060724_

Round 1

Reviewer 1 Report

In this manuscript, the chemical compositions of the essential oils from the aerial parts and bulbs of two cultivars of Allium sativum named 'Bianco del Veneto' and 'Staravec'. In addition, this study provided a deep insight on the antibiofilm activity of the above mentioned essential oils, which worth reading. I will recommend this manuscript after minor revision.

  1. In Title the word ‘leaves’ was used, however the words ‘aerial parts’ were frequently encountered in the main text not ‘leaves’.
  2. As indicated in Table 1, diallyl disulfide is another main component of the essential oils. Therefore, it is better to add this compound in the Abstract.
  3. Please define the abbreviation ‘EO’ as ‘essential oil’ when it appeared for the first time.
  4. The positive controls were missed in Tables 3 and 4.
  5. As indicated in Tables 3 and 4, the inhibitory activity on the biofilm formation and the cell metabolism were tested in two different concentrations, it is better to add the concentration after the data when referred in the main text, such as Page 5 Lines 130, 142 and 149, Page 6 Lines 180 and 184, Page 7 Lines 195 and 206.
  6. As mentioned in the references, both allicin and diallyl disulfide exhibited antibacterial acticities. However, for the EOs from the bubls of 'Staravec', Table 3 showed no inhibitory on the biofilm formation of all the four pathogenic bacterial strains. It is difficult to understand.
  7. As indicated in Table 1, both allicin and diallyl disulfide are the main components of the EOs from the bubls of 'Bianco del Veneto' and 'Staravec'. And the ratios of them are similar. However, the EOs from the latter exhibited good inhibitory activity on the cell metabolism of E. coli, whereas negative for the EOs from the former as indicated in Table 4. Why?
  8. Other corrections as followings:

Page 2 Line 64: ‘Sars-Cov-2’  →  ‘SARS-CoV-2’

Page 2 Line 68: italic ‘Allium’  →  ‘Allium

Page 2 Line 90: ‘2,2-Bis(ethylthio)-’  →  ‘2,2-bis(ethylthio)-’

Page 2 Line 91: italic ‘Z’  →  ‘Z

Table 1: italic ‘Z’  →  ‘Z’    ‘E’  →  ‘E

Tables 2–4: The species names should be written in italic type.

Author Response

  1. In Title the word ‘leaves’ was used, however the words ‘aerial parts’ were frequently encountered in the main text not ‘leaves’. Thank you for your observation. We changed the term “leaves” in the title with “aerial parts”
  2. As indicated in Table 1, diallyl disulfide is another main component of the essential oils. Therefore, it is better to add this compound in the Abstract. Thank you for your observation. We changed the abstract section.
  3. Please define the abbreviation ‘EO’ as ‘essential oil’ when it appeared for the first time. Thank you for your observation. We modified accordingly.
  4. The positive controls were missed in Tables 3 and 4. Thank you for your observation. Ther was an error in the caption of the tables 3 and 4. All calculation, in fact, were done considering the untreated cells. We changed the caption to tables 3 and 4. Sorry for the misunderstanding.
  5. As indicated in Tables 3 and 4, the inhibitory activity on the biofilm formation and the cell metabolism were tested in two different concentrations, it is better to add the concentration after the data when referred in the main text, such as Page 5 Lines 130142 and 149Page 6 Lines 180 and 184, Page 7 Lines 195 and 206. Thank you for your observation. We modified page 5 lines 130, indicating, ath the beginning of sentence, that such results were referred to the use of 20 microliters/ml. We also indicated that the experiments were performed using 10 and 20 microliters/ml in the caption of the table 3 and 4; we also added the concentration in the other points you suggested. However, on line 142, the percentages were referred to the singular compounds, so in that case we didn’t make it;
  6. As mentioned in the references, both allicin and diallyl disulfide exhibited antibacterial acticities. However, for the EOs from the bubls of 'Staravec', Table 3 showed no inhibitory on the biofilm formation of all the four pathogenic bacterial strains. It is difficult to understand.
  7. As indicated in Table 1, both allicin and diallyl disulfide are the main components of the EOs from the bubls of 'Bianco del Veneto' and 'Staravec'. And the ratios of them are similar. However, the EOs from the latter exhibited good inhibitory activity on the cell metabolism of E. coli, whereas negative for the EOs from the former as indicated in Table 4. Why? Thank you for your observation.  We have changed these two paragraphs. We provided to calculate the coefficient of correlation taking into account the chemical composition, and the action exhibited both by the bulbs and aerial parts of the EOs of “Bianco del Veneto” and “Staravec”, trying to better explain our hypotheses and consideration.
  8. Other corrections as followings:

Page 2 Line 64: ‘Sars-Cov-2’  →  ‘SARS-CoV-2’. Thank you. We modified accordingly

Page 2 Line 68: italic ‘Allium’  →  ‘Allium’. Thank you. We modified accordingly

Page 2 Line 90: ‘2,2-Bis(ethylthio)-’  →  ‘2,2-bis(ethylthio)-’ Thank you. We modified accordingly

Page 2 Line 91: italic ‘Z’  →  ‘Z’. Thank you. We modified accordingly

Table 1: italic ‘Z’  →  ‘Z’    ‘E’  →  ‘E’  Thank you. We modified accordingly

Tables 2–4: The species names should be written in italic type. Thank you for your observation. We modified accordingly

Reviewer 2 Report

See attached file

Author Response

The authors presented a study on the chemical composition of essential oils from different garlic cultivars and plant parts. The biofilm-inhibiting and anti-metabolism action of the essential oil was investigated on different microorganisms. The experimental work of the study is complete and systematic, as the composition of the extracts can be directly linked to the antibacterial activity of the essential oils. However, at this point, the discussion of the results should be much more in detail and elaborated. More specifically, there is a lot of literature available on the antibacterial action of essential oils from different garlic cultivars, even a similar study (using the same methods) from the authors’ research group earlier this year. Results should be compared to (at least try to) converge to a conclusion on which cultivars (and plant parts) would be most suitable for which applications. Thank you for your observation. We tried to enrich the manuscript following your right comments.

 Specific comments: • L54-55: …stronger evolutionary drug resistance due to the careless use …

  • L56-58: Is there really an increasing capacity of bacteria to form biofilms? Maybe researchers have become more aware of biofilms as a challenge, but they are not more prevalent than before? Please confirm the statement using the scientific literature and adapt if necessary. Thank you for your right comment. We have changed the sentence.
  • L120-122: Can the authors clarify why concentrations lower than the MIC were selected? If the MIC was tested on planktonic cells, one would expect the necessary inhibiting concentrations to be higher for biofilms. Thank you for your observation, We have considered two concentration lower than those capable to inhibit the bacterial growth, so that bacteria, at lower concentration of the EOs tested, can have still the capacity to grow and form their biofilm.
  • L150: … seemed to…we changed
  • L153: It is strongly put that this breaking effect on biofilm inhibition is due to the allicin. With the big differences in composition between the different extracts, it’s difficult to say that allicin is responsible for the lower biofilm inhibition potential. Thank you for your comment. We modified the discussion
  • Tables 2-4: Statistical analysis was always conducted in comparison to Tetracycline as a control. However, the reasoning behind this has not been clarified. Neither has tetracycline been mentioned anywhere in the materials and methods section. o Please add information on this methodology and reasoning in the materials and methods section. Please add discussion on the comparison with tetracycline in the results and discussion section. Thank you for your observation. we modified the discussion.
  • There must be a use of comparing with the antibiotic. There was an error in the caption of table 3 and 4. In fact, we considered as positive control the untreated cells, for whose the inhbitory action was considered 0. Thus we provided to correct the captions both in Table 3 and Table 4.
  • • L192-195: Please adapt the sentence to: The essential oils, when used at the highest concentration, always exerted an inhibitory force on the bacterial metabolism of A. baumannii, with the ‘Staravec’ EOs being more effective than the ‘Bianco del Veneto’ ones, especially for the essential oil recovered from the bulbs (57.79% and 24.10%, respectively). Thank you for your help, we changed.
  • L200-206: It is not completely clear what the authors mean in this paragraph. Is it meant that, even though biofilm inhibition is not dependent on allicin concentration, the antimetabolic action is? This part of the text should be made more clear. Additionally, with the very diverse composition of the different extracts, how can the authors be sure that any difference in observed effects is solely due to differences in allicin concentration? Thank you for your observation. Of course you are right, the differences in the behaviour exhibited by the EOs cannot be attribuible merely to allicin. We provided to calculate the coefficients of correlation taking into account the results of table 1 and 3. And, in more cases, our hypotheses could be reinforced by the analysis of correlation coefficients. we have observed more than once that the behavior exhibited by an EO or a compound on the ability to inhibit the biofilm may be different from the same exhibited on the metabolism of bacterial cells present within the biofilm, but we believe this is normal, because, as we know, the action of an EO, or a compound, or a natural extract against a pathogen can be carried out in different ways, which may affect their ability to influence the metabolism of the cells within the biofilm, but also that to cause damage to their cell wall etc.• Therefore, in the specific case, if we observed an inhibitory activity on the formation of the biofilm and this did not correspond - in some cases - to a similar ability of the EOs to act on the metabolism, other mechanisms probably came into play.

 L214-215: Is this statement based on experimental findings in the current study or on literature? Please clarify and add references if necessary. Thank you for your comment. We added some references corroborating our hypothesis

  • L216-218: The authors here refer to two previous publications from their research group. In one of them, they have investigated the biofilm inhibition and metabolic action of two other garlic cultivars than the ones tested in this study. Since the same microorganisms were tested, and the same methods were used, the results should be much more elaborately compared than the brief description in this sentence. Thank you for your observation. We changed the section and, taking into consideration the data published by Polito et al, we calculated the coefficients of correlations trying to explain the influence of singular compounds on the inhbitory capacity of the EOs recovered by those two cultivars, “Rosso di Sulmona” and “Rosso di Spagna”, where it was confirmed the negative effect, almost generally, exhibited by allicin and the positive effect exhibited by other sulfide compounds, such as diallyl sulfide and propyl allyl sulfide. This was done both considering the biofilm inhibitory capacity of those EOs and their capacity to inhibit the metabolism of the bacterial cells within the biofilm. Now these are not more L 216-218 , because we enriched the manuscript also in the introduction section
  • L216-227: The authors should make a much more detailed and elaborate discussion on how the EO of different Garlic cultivars (taking into account also the different parts from the same plant) affect the different pathogens (biofilm inhibition and metabolism). The discussion should be much more detailed than the rather vague descriptions which are offered in this paragraph. If not at this location, it can also be done at another location in the manuscript. Looking at the referred literature, there have been quite some studies on different cultivars in different studies, even quite a few from the authors’ research group. It should not be the goal to just keep on testing new cultivars in every study without at least trying to converge to a most suitable option (with the current available knowledge). Some info on which cultivars (and plant parts) would be best used for which applications should also be added, preferably using the biofilm inhibition potential and metabolism action against different pathogens as argument. Thank you very much for your comment that helped us to meliorate (we hope) the manuscript. We provided to improve the discussion.
  • L274: transfer line •

 L289-293: Was the MIC calculated for planktonic bacterial cells or biofilms? Please add more info on this. The test with resazurin was performed on planktonic cells. MTT was performed on cells enclosed in the biofilm, after having discarded the non adhered cells

Reviewer 3 Report

The antibacterial activity of Allium sativum is well known. Here, Nazzaro and colleagues should better emphasized the added value of their results. They should also reconsider some result interpretation.

Detailed comments:

 1.As various scientific publications have already described the antimicrobial activity of this plant, the authors should better emphasized the added value of their research/results. An assessment of the diversity of the activity/chemical composition in function of the plant characteristics could it be the central axis? In that sense, comparison between previous publication data and their observations (even presented in Tables) could be presented.

2. Resasurin assay and MTT are both giving an assessment of metabolically active cells. So, it is illogical to use one as a viability marker (resazurin in the MIC assay) and the other one as an indicator of metabolism (MTT). Therefore, the author should reconsider their result interpretation and/or methods

3. Crystal violet staining is a biomass staining, giving mostly semi-quantitative assessment (% of biomass = how many living bacteria?). Therefore what would mean for example a "reduction of 26,31%" as reported, as an antimicrobial agent? I recommend to count in logarithmic reduction of colony forming unit. Usually, we expect to have at least 2 log reduction meaning an activity of 99% inhibition for an efficient antimicrobial/antibiofilm compound. The lowest reported activity is a 50% reduction (this is with the resazurin assay for MIC determination).

4. Reference 1 is not scientifically relevant compared to existing scientific publications

5. the activity of A. sativum on SARS-CoV-2 is based on two publications which are speculative (in silico analysis for da Silva and colleagues and review without scientific experiments for Alaba and colleagues), not based on relevant microbiological assays

Author Response

The antibacterial activity of Allium sativum is well known. Here, Nazzaro and colleagues should better emphasized the added value of their results. They should also reconsider some result interpretation.

Detailed comments:

 1.As various scientific publications have already described the antimicrobial activity of this plant, the authors should better emphasized the added value of their research/results. An assessment of the diversity of the activity/chemical composition in function of the plant characteristics could it be the central axis? In that sense, comparison between previous publication data and their observations (even presented in Tables) could be presented. Thank you for your right comment. We have tried, in this revised version of the paper, to improve the discussion.

  1. Resasurin assay and MTT are both giving an assessment of metabolically active cells. So, it is illogical to use one as a viability marker (resazurin in the MIC assay) and the other one as an indicator of metabolism (MTT). Therefore, the author should reconsider their result interpretation and/or methods. We have used resazurin to identify the concentration of EOs capable to inhibit the growth of the planktonic cells. MTT is used to study the metabolism of the sessile cells present within the biofilm, after that the planktonic cells, non adhered to the surface to microplate, were discarded and biofilm washed to avoid their presence. MTT test was performed using two concentrations of EOs lower than those identified through the resazurin test.
  2. Crystal violet staining is a biomass staining, giving mostly semi-quantitative assessment (% of biomass = how many living bacteria?). Therefore what would mean for example a "reduction of 26,31%" as reported, as an antimicrobial agent? We mean that, respect to the control, where we assumed an inhibitory activity= 0, the EO, giving rise to a reduction of 26.31 % , caused a reduction of the biofilm capacity of that strain, thus it, in our opinion, has, by the whole, an antibacterial activity. I recommend to count in logarithmic reduction of colony forming unit. Usually, we expect to have at least 2 log reduction meaning an activity of 99% inhibition for an efficient antimicrobial/antibiofilm compound. The lowest reported activity is a 50% reduction (this is with the resazurin assay for MIC determination). We have used different times the crystal violet staining test for the evaluation of the capacity of different compounds to act on the biofilm capacity given by bacteria. These are some of our previous works, in last two years, reporting the test: F Polito et al. Chemical Composition and Agronomic Traits of Allium sativum and Allium ampeloprasum Leaves and Bulbs and Their Action against Listeria monocytogenes and Other Food Pathogens, Foods 11 (7), 995, 2022; F.Fratianni et al. Fatty Acid Composition, Antioxidant, and in vitro Anti-inflammatory Activity of Five Cold-Pressed Prunus Seed Oils, and Their Anti-biofilm Effect Against Pathogenic Bacteria. Frontiers in Nutrition 8 (fnut.2021.775751), 2021; L. De Martino et al. Variations in composition and bioactivity of Ocimum basilicum cv ‘Aroma 2’essential oils. Industrial Crops and Products, 2021; F.Nazzaro et al. Anti-biofilm properties exhibited by different types of monofloral honey.Multidisciplinary Digital Publishing Institute Proceedings 66 (1), 16,2021; F.Nazzaro et al. Antibiofilm properties exhibited by the prickly pear (Opuntia ficus-indica) seed oil. Multidisciplinary Digital Publishing Institute Proceedings 66 (1), 29, 2021. Thus, we have used the same method, already approved in our previous works. With crystal violet it should be difficult to perform a count of colony forming units, because the treatment with alcohol to fix biofilm  let to the cell damage and death. However, in the future, we can perform a concurrent test to evaluate the % of living bacteria, using the method you propose. Thank you very much for your suggestion.
  3. Reference 1 is not scientifically relevant compared to existing scientific publications. Thank you. We provided to delete it from the text and reference list
  4. the activity of A. sativum on SARS-CoV-2 is based on two publications which are speculative (in silico analysis for da Silva and colleagues and review without scientific experiments for Alaba and colleagues), not based on relevant microbiological assays. Thank you for your comment. We have modified the sentence, indicating that those are in silico studies that should be corroborated by in vivo tests.

Round 2

Reviewer 2 Report

The authors have addressed all comments. The manuscript can be accepted in its current form.